# Comparison of Audience Behavior between eSports and Authentic Sports Fans

**DOI:** 10.3390/bs14040313

**Published:** 2024-04-11

**Authors:** Taeyeon Oh, Joon-Ho Kang, Younghan Lee, Soyon Michelle Choi

**Affiliations:** 1Seoul AI School, aSSIST University, Seoul 03767, Republic of Korea; tyoh@assist.ac.kr; 2Department of Kinesiology, Seoul National University, Seoul 08826, Republic of Korea; 3Department of Kinesiology, Mississippi State University, Starkville, MS 39762, USA; yl690@msstate.edu; 4Stan Richards School of Advertising and Public Relations, Moody College of Communication, The University of Texas at Austin, Austin, TX 78712, USA; m.soyonchoi@utexas.edu

**Keywords:** esports, authentic sports, comparison, use and gratification, media transportation

## Abstract

This study analyzed how the behavioral patterns of esports and authentic sports viewers differ, adopting user and gratification theory and media transportation theory. In particular, it was investigated whether there was a difference in behavioral patterns according to the experience of playing the sport even among authentic sports viewers. As a result of analyzing the relationship between viewers’ motivation and media transportation outcomes through structural equation modeling and multigroup structural equation modeling, it was observed that cognitive motivation was more important for esports viewers than for authentic sports viewers. A second analysis of comparisons among fans of authentic sports showed that viewers with sports experience had greater cognitive needs. This result shows that there is a difference between the viewer behaviors of esports and traditional sports, but it is concluded that the presence or absence of sports participation experience rather than content is the factor that separates the difference.

## 1. Introduction

There are various discussions on whether electronic sports (esports) is a developed form of sports or another type of industry that borrows the name of sports [1,2]. There is a view that esports are not sports because it involves extremely limited physical activity [1]. In contrast, there is an opinion that it is appropriate to consider it a sport in that muscle cooperation is made possible by judging visual information within a short window of time [3]. Aside from these fundamental discussions, the structure of the esports industry is very similar to that of the existing professional sports industry because the former has evolved by benchmarking the latter [4]. Despite these similarities, only a few studies have focused on the behavior of esports consumers. The similarities and differences between esports and general sports consumers in terms of behaviors have not been examined in detail. One major difference between esports consumers and fans of sports in general is that every esports fan is a gamer [5]. Some fans of authentic sports (e.g., baseball), have never played the sport. However, in the case of esports fans, most of them have experienced the sport as gamers first, and this experience leads to esports viewing.

Most studies on esports consumption motivations have derived esports-specific factors, developed related measurement tools, and analyzed the behavior of esports consumers based on them [6,7,8]. Researchers measured the motivational factors of esports consumers by applying 15 motivational factors from existing sports spectatorship motivation scales and comparing esports consumers with authentic sports consumers [4]. They measured the motivation factor with single-questionnaire items, and consumer behavior resulting from motivations was simply compared with the frequency of viewing. Therefore, their results had limitations. However, the factors related to the consumption motive developed in other esports studies include many factors that are unique to esports. Thus, they are unsuitable for comparative studies through application to general sports consumers.

The current study examines whether there are similarities or differences between the two viewer groups, with a focus on general media theories. It goes beyond the study of fan behavior based on authentic sports or esports motives. Although there are various studies on viewing motives, most recent multichannel and new-media-environment viewing motivations have been explained by Use and Gratification Theory (UGT). UGT suggests that consumers have needs that they want to gratify and make viewing decisions in a way that satisfies these needs [9]. The advantage of UGT is that different audience groups can be investigated through the same motivating factor, which makes it easy to compare different groups [9]. In the literature on media audience behavior, several studies have examined whether positive viewing experiences were achieved through the psychological state of viewers and through viewing time or ratings as a result of media viewing motives. Flow theory, especially transportation theory, is an excellent indicator of viewers’ psychological experience as it measures the extent to which they are immersed in their viewing experience [10].

This study analyzes the similarities and differences between the viewing motivation and outcome behaviors of esports and authentic sports viewers by adopting UGT and transportation theory. Most esports fans are gamers, and gaming leads to esports consumption. Authentic sports spectatorship rarely involves real play. Among sports consumers, there may be differences between viewers who actually play the sport and those who do not. In the current study, we compare the behaviors of these players with those of non-players and analyze the differences in comparison with esports viewers. Finally, we compare the behaviors of esports viewers with those of authentic sports viewers.

## 2. Literature Review

### 2.1. Electronic Sports or Sports

The term ‘esports’ has been used widely in many fields, although its boundaries remain undefined. It refers to the events of organized video gaming competitions and sometimes means video games [11]. However, in the sports literature, video gaming as a spectator sport is generally considered an esport [7,12]. Systematically well-organized video gaming competitions, including professional players, teams, and governing bodies (league offices, associations, etc.) are considered the subject of esports studies.

Several studies on the comparison between authentic sports and esports have focused on the differences that generally center on the involvement of physical activity [13]. However, esports and authentic sports are hardly distinct when seen from an event standpoint. Esports comprise leagues and events that benchmark the structure of professional sports like Major League Baseball or the Champions League in UEFA. Therefore, given the similarity in the organizational structures, traditional sports organizations actively engage in forming esports organizations (e.g., the esports team PSG Talons under the Paris Saint-Germain Soccer team). Traditional media outlets also broadcast esports. In terms of broadcasting spectatorship, there is little difference between authentic sports and esports [4].

From this perspective, one may expect the attitudes of fans to be a major difference between esports and authentic sports. In general, most esports fans are gamers and have experience playing video games [5]. In particular, a large number of esports viewers play the game currently. Therefore, many esports fans watch esports to improve their gaming skills and obtain new information along with the various experiential pleasures they obtain while watching the game [6].

If we compare the literature on motives for viewing, this difference in attitude can be confirmed clearly. For example, Trail and James [14] reported that the motives for being a spectator include the aesthetic qualities of the event, display of aggression, betting, and dramatic aspects. These factors pertain to the emotional fulfillment that can be obtained while watching a television show. According to Kim and Mao [15], emotional relief, sociability, and storytelling are motivations for the consumption of mediated sports. Esports motivation studies have identified cognitive factors, namely information seeking and learning, as part of spectatorship motivations [7,8,16,17]. Esports viewers try to improve their gaming skills, learn new strategies, and discover updated game trends (or metagames) by watching.

Studies on the motivation for watching authentic sports include acquiring knowledge through watching as a factor. However, motivational factors in existing sports cannot be considered the same as those for esports. Most studies on esports viewing motivation have focused on developing new measurement tools instead of using existing ones [12]. According to Reitz and Hallmann [12], most consumer behavior research on esports considers it an independent study area rather than a part of sports. Thus, there is a limited number of studies comparing the motivations and behavioral outcomes of authentic sports and esports viewers [4,18]. However, Brown and colleagues [18] compared the effects of the esports participation experience on esports and authentic sports viewing motivation; thus, it cannot be said that they compared different consumer groups. Other researchers [4] examined the differences between esports and authentic sports consumption motivations; however, it is about attendance motivation and not media-consumption-related factors.

Studies on viewing motivation or related behavioral outcomes between authentic and esports viewers are scarce. As sports-related motivators are traditionally used in the sports-specific literature, there are cases where these measurements are not interchangeable with esports-related content. However, if we change our perspective and look at sports media from the perspective of media research, we can assume that similar motivating factors of authentic sports viewers drive esports fans. This can be explained by the UGT in media studies [9].

### 2.2. UGT

UGT research originated from analyzing mass communication media audiences (Katz et al., 1974 [19]). The main concerns of UGT are the psychological and sociological factors that form needs and the interactions between these needs and gratification. UGT has been in the spotlight more significantly in the 21st century, where numerous cable networks that were launched have broadened viewers’ choices, and the limitations of time and space have disappeared with the development of new media through the Internet [9,20]. UGT assumes that viewers watch or choose media to seek gratification [19,20,21]. Therefore, it is possible to link the gratification sought by audiences as a motivating factor through UGT [22]. UGT can be applied widely to explain various consumption behaviors related to media consumption. Scholars have adopted UGT to explain the motivation for choosing a television channel or program, such as a sports show [19,20,21,23], and to investigate the usage of media devices like mobile phones [24]. User motivations online or on social network services (SNSs) can be explained through UGT [25,26,27]. Studies have explained the user behavior of social live-streaming services adopting UGT [6,16]. Sports and esports-related media studies have also benefited from using UGT. According to Rietz and Hallmann [12], over half the esports audience behavioral studies have adopted UGT to explain the mechanism underlying user motivations. Authentic sports viewership was also explained by UGT motivations [23,28].

Different motivations were derived in UGT studies. However, West and Turner [22] suggested five need types that are representative: cognitive, affective, personal integrative, socially integrative, and tension release needs. Hamari and Sjöblom [6] used these five needs to explain esports streaming service user motivations. Affective and cognitive needs are the enjoyment and information and game strategy that users seek, respectively. Social integrative needs include companionship and feelings of connectedness among esports viewers. Tension release involves a feeling of escape or relaxation by watching esports. Personal integrative needs are connected to the desire to be recognized by other users. These factors explain the behavior of esports streaming service users and can be applied to the general esports viewership behavior. As the personal integrated need is the desire to be recognized by other fans or content providers [6], it is not appropriate for the current study. Therefore, we adopted only four motivations.

### 2.3. Media Transportation

Researchers have adopted diverse outcome variables owing to audience motivations. Among them, one of the commonly used variables was watching hours. It measures the amount of time the audience spends watching a sporting event [6]. Various psychological changes that take place owing to various viewings are measured and used as outcome variables, such as satisfaction, enjoyment, and flow. Among these, media transport enables an interesting point of view in examining the results according to the viewing motive [10].

Transportation theory explains the status wherein the audience is transported into a narrative and temporarily loses access to the real world [29]. This is a special case in which the audience is completely immersed, meaning that the audience in this state completely moves to the virtual reality provided by the media and is separated from reality [30]. Transportation is a proxy for the enjoyment or satisfaction that media audiences feel while watching and can ultimately lead to a change in viewers’ attitudes toward the media context [10]. Transportation is suitable for use as an outcome variable according to the viewing motive of the media audience. It comprises experience and attention [10]. Experience refers to the degree to which the audience has been immersed in media content while watching, and attention refers to the degree of forgetting about the real world. A research model can be proposed by combining these UGT motivation and transportation factors. A proposed path model is presented in Figure 1.

### 2.4. Research Model and Hypotheses

Esports and authentic sports may be assumed to have a common purpose, which is to obtain enjoyment from watching as a media audience. However, the cognitive needs to acquire information may differ between esports and authentic sports viewers because the former are mostly gamers who play games, whereas the latter are not necessarily players. However, we can think that a person who plays a sport would like to obtain information on playing it better while watching it. They will exhibit a behavior similar to that of esports viewers. Against this background, this study established two research hypotheses on the factors that motivate viewing. Based on the proposed path model, the current study suggests the following hypotheses:

**H1.** *The cognitive needs of esports viewers are higher than those of authentic sports viewers*.

**H2.** *The cognitive needs of sports player audiences are higher than those of non-player ones*.

**H3.** *The path relationship of esports viewers is different from that of authentic sports viewers*.

**H4.** *The authentic sports viewers’ path relationship is moderated by playing experience*.

**H5.** *The authentic sports players’ path relationship is similar to that of esports viewers*.

## 3. Method

### 3.1. Data Collection and Measurement Scales

Samples were collected randomly through an online panel in both the US and South Korea. We collected data through random sampling of people who passed the screening questions “Are you a sports fan?”/“Are you an esports fan?” from a response panel maintained by research firms in the United States and Korea. This study was approved by the University of Mississippi’s Institutional Research Board under protocol number 22x-221. A total of 628 responses were received, and 556 data lines were used, excluding insincere answers. The respondents removed were mostly those who answered all questions with either 1 or 7, and we also excluded those who responded insincerely to demographic data. Among the 556 participants, 223 and 333 were esports and authentic sports audiences, respectively. Among the 333 authentic sports viewers, 130 had never played the sports they watched, and 203 had experience playing the sports or were playing it at the time of study. The participants were on average 40.34 years of age (S.D. = 14.28), and the average age of the esports and authentic sports viewers was 32.05 (S.D. = 8.59) and 45.89 (S.D. = 14.65), respectively.

We measured four UGT motivations, media transportation factors, and weekly watching hours. The four UGT motivations (affective, cognitive, social integrative, and tension release needs) were adopted from Hamari and Sjöblom’s [6] esports streaming service viewing motivations. Two media transportation factors (experience and attention) were measured using the items suggested by Tal-Or and Cohen [10]. The questionnaires were modified to suit the esports and authentic sports contexts. Each item was measured using a 7-point Likert scale anchored by (1) strongly disagree to (7) strongly agree. Weekly hours of watching were investigated through an open-ended question to fill out the exact watching hours. The measurement items are presented in Table 1.

### 3.2. Data Analyses

To verify the validity and reliability of the measurement, we conducted a CFA and derived Cronbach’s alpha. First, two hypotheses were tested through a *t*-test, and the others were verified through structural equation modeling (SEM). Multigroup SEM was applied to understand the difference between esports and authentic sports viewers and players and non-players among sports viewers. Before multigroup SEM, the invariance test for the measurement and structure was conducted by comparing unrestricted and weakly (factor loadings only), strongly (factor loadings and intercepts), and strictly (factor loadings, intercepts, and residuals) restricted CFA results.

## 4. Results

### 4.1. Measurement Validity and Reliability

Before testing the hypotheses, we conducted a CFA and related tests to verify the measurement reliability and validity. The model fit indices showed acceptable results, which guaranteed that the data fit the hypothesized measurement model (see Table 1). Both the comparative fit index (CFI) and Tucker–Lewis index (TLI) exceeded 0.9, the root mean square error of approximation (RMSEA) was 0.070, and the standard root mean square residuals (SRMRs) were 0.063; all were within the acceptable range of the goodness-of-fit as suggested by Hair et al. [31]. The Cronbach’s alpha, composite reliability (CR), and average variance extracted (AVE) were derived to verify the internal consistency and discriminant validity, and the indices exceeded the threshold (0.7, 0.7, and 0.5, respectively; see Table 2). The correlations of the variables are presented in Table 2, and the value of most of the correlations was not larger than the square root of the AVE, except for the correlation between outcome variables (experience and attention). However, all values were no larger than the criterion of variable discrimination (0.85, [32]). Therefore, we can confirm the measurement validity and reliability.

### 4.2. Results of Analyses

After validating the measurements, we conducted a path analysis for the data. The proposed path model showed acceptable model fit indices (x^2^ = 428.827, df = 115, *p* < 0.001, CFI = 0.946, TLI = 0.928, RMSEA = 0.070, SRMR = 0.063). We could not find any relationship between UGT motivation factors and weekly watching hours. However, the affective, social, and tension release factors were significantly related to the experience, and all four motivations impacted attention (see Table 3). Thus, we can conclude that whereas watching hours were not directly related to UGT motivations, media transportation was influenced by them.

To understand the difference between esports and authentic sports viewers, we conducted a *t*-test for the UGT motivations before the multigroup SEM analyses. The social factor was not different from esports and authentic sports viewers, but the other three factors showed significant differences. Esports viewers showed higher cognitive motivations, and authentic sports viewers had higher affective and tension-release motivations (Table 4).

For the test between non-players and players among esports viewers, players showed higher cognitive and social motivations (see Table 5). Thus, esports viewers had a stronger cognitive desire, namely the motivation to obtain information, than authentic sports viewers. A similar result was also observed for players when compared to authentic sports viewers. These results supported the first and second hypotheses.

We used multigroup SEM to verify the difference between esports and authentic sports viewers to gain a deeper understanding. We compared the chi-square indices for the unrestricted, weakly, strongly, and strictly restricted models to verify the model invariance. The multigroup path analysis was available at least only when the invariance between the unrestricted and weakly restricted model was verified [33,34]. The weakly restricted model was invariant with the unrestricted model (∆x^2^ = 13.541, df = 11, *p* = 0.259) whereas the strongly (∆x^2^ = 41.146, df = 11, *p* < 0.001) and strictly restricted ones (∆x^2^ = 90.044, df = 17, *p* ≤ 0.001) did not show invariance (see Table 6).

Through the multigroup path analysis, we found that the esports and authentic sports viewers showed different paths. In the case of the former, social (β = 0.625, *p* < 0.05) and tension-relief needs (β = 0.378, *p* < 0.05) were significant among the paths toward experience, and cognitive (β = 0.300, *p* < 0.10), social (β = 0.283, *p* < 0.05), and tension-relief (β = 0.445, *p* < 0.05) needs were significant in the path toward attention. In the case of authentic sports viewers, the affective (β = 0.153, *p* < 0.05), social (β = 0.589, *p* < 0.05), and tension-relief (β = 0.208, *p* < 0.05) needs were significant for the path toward experience and attention (β = 0.349, β = 0.418, β = 0.316, and *p* < 0.05, respectively). Affective (β = 1.197, *p* < 0.10) and social factors (β = 1.323, *p* < 0.10) affected watching hours significantly. The effects of social and tension-relief factors on media transport were similar in both groups, but affective factors worked only for authentic sports viewers and cognitive factors worked only for esports viewers (see Table 7). This supports Hypothesis 3.

We verified the multigroup invariance for the non-players and players among authentic sports viewers (see Table 8).

The multigroup path results are presented in Table 9. We observed significant paths from affective (β = 0.387, *p* < 0.05) and social (β = 0.514, *p* < 0.05) factors to experience, and from affective (β = 0.499, *p* < 0.05), social (β = 0.500, *p* < 0.05), and tension-relief (β = 0.341, *p* < 0.05) factors to attention in non-players. For the players, the cognitive factor (β = 0.193, *p* < 0.10) was connected to experience instead of the affective factor, and tension-relief (β = 0.658, *p* < 0.05) and social needs (β = 0.304, *p* < 0.05) were significantly related. All four needs were significantly related to attention (see Table 9). Watching hours were influenced by affective needs (β = 1.818, *p* < 0.10) for non-players and social needs (β = 1.814, *p* < 0.10) for players. Even among authentic sports viewers, the viewers who play sports had different effects on the viewing motivation and results than those who did not. These characteristics appear similar to those of esports viewers. These results support hypotheses 4 and 5.

## 5. Discussion

Through an empirical analysis, this study confirmed how the relationship between esports and authentic sports audiences’ viewing motives and related outcomes are different. Cognitive factors play an important role for esports viewers, whereas affective ones play an important role for authentic sports viewers. However, among the latter, players’ characteristics were similar to those of esports viewers.

This study makes a notable contribution to the literature. Pizzo et al. [4] examined the audience that visited a stadium and found that there was a significant difference between vicarious achievement and family bonding vis-à-vis the difference in motivation factors between esports and authentic sports. However, the current study analyzed the motives of viewers and found that esports viewers consider it important to obtain information as both viewers and gamers. Unlike studies on the motivation of esports viewers [7,8,16,17], this study analyzed authentic sports viewers and esports fans to explore differences in their behaviors.

The cognitive factor was key in distinguishing the behavioral patterns of viewers. Cognitive or information-seeking needs were found in esports viewers. Improving their game skills and acquiring new game tactics are important motivations for esports viewers [35]. One of the main motivations for watching video game streaming online is to learn new game meta and gain information that can help improve one’s gaming skills [16,17]. When compared with these previous research results, this study can also derive consistent conclusions.

The current study analyzed whether the effect of cognitive needs is a characteristic effect of esports content or whether there is a difference as a result of actual sports participation. When analyzed based on whether or not authentic sports viewers play the sports they watch, it was found that information search was an important factor for players, on the lines of esports viewers, but had no effect on non-players. This supports the hypothesis that esports viewers will exhibit differences in their viewing behavior from those of existing sports viewers as most of them are gamers. Authentic sports viewers confirmed that players had different viewing motives from non-players. This result emerged owing to the difference in the viewer’s experience or participation in the game rather than the difference in content between esports and authentic sports. The magnitude of the impact exerted by cognitive factors on influence yields intriguing outcomes. It is observed that cognitive effects on attention are more pronounced among esports viewers compared to those engaged in authentic sports. This suggests that cognitive factors may exert a more significant influence on viewers, primarily because esports offer a greater potential for applying the information acquired during viewing to actual gameplay.

The effect of watching hours per week on motivators was not found in most cases. This is inconsistent with the results of many studies that use weekly viewing time as a dependent variable [7,8,16,17]. None of the motivating factors affected viewing time for esports viewers. However, in authentic sports, emotional and social factors had a positive effect on viewing time [17]. First, motivational factors did not affect the viewing of esports, which contradicts the behavior of existing esports streaming viewers. We can hypothesize that, unlike authentic sports, we failed to measure the esports viewers’ viewing time consistently as the esports season’s game weeks are short and the duration is different for each game. Therefore, the answer to the simple average viewing time may vary greatly based on the game the audience enjoys. The viewing time of esports viewers, who mainly watch online or through mobile devices, may not be properly measured because the same environment as that of general TV viewers is assumed.

The results of the current study contribute in the following ways. Research has tended to treat esports as similar to authentic sports in terms of markets but as different in terms of content. This was articulated in Reitman et al. [11]. Many esports studies have developed tools to measure consumer motivation, which assumes that esports differs from existing sports in terms of content. However, this study shows that the factor that causes a difference in viewing behavior is the viewer’s experience rather than the content itself. Players among authentic sports viewers behaved like esports viewers. Within the same sports audience, reactions differed based on the presence or absence of experience. This study thus confirmed how consumer behavior changes from a behavioral point of view while breaking away from the conventional content–conceptual point of view in esports research. This marks a change in perspective from previous studies.

## 6. Conclusions

The difference between the behavioral patterns of esports and typical sports viewers was analyzed through a questionnaire. This difference was marked by experience participating in sports. In particular, it was observed that cognitive motivation was more important for esports viewers than for authentic sports viewers. A second analysis of comparisons among fans of authentic sports showed that viewers with sports experience had greater cognitive needs. This result shows that there is a difference between the viewer behaviors of esports and traditional sports, but it is concluded that the presence or absence of sports participation experience rather than content is the factor that separates the difference.

This study has a few limitations. First, research samples were collected through random sampling. Random sampling can sometimes fail to represent the diversity within the population, leading to sample bias and a lack of generalizability of the findings. In addition, owing to differences in the viewer demographics, it was inevitable to collect samples from the same population group. Esports viewers mainly use Internet-oriented streaming services rather than TV, so it is not true that the actual viewing time or behavior is entirely the same as that of TV viewers. However, to compare both groups, it was assumed that they were in similar environments. While watching media on a mobile device is different from TV, viewing specific content with a purpose is similar. Future studies must control the viewing device with due consideration for this point. The desired conclusion could not be drawn because the viewing time could not be identified by analyzing the viewer’s viewing record, and the average viewing time per week was based on memory. In the future, systematic research must rely on more specific data vis-à-vis viewers’ viewing time. Second, this study attempted to collect an equal number of samples for both groups, but many esports-related respondents responded insincerely, so the current study was conducted with an unbalanced sample. In the future, more valid studies should be conducted through more accurate sampling.

## Figures and Tables

**Figure 1 behavsci-14-00313-f001:**
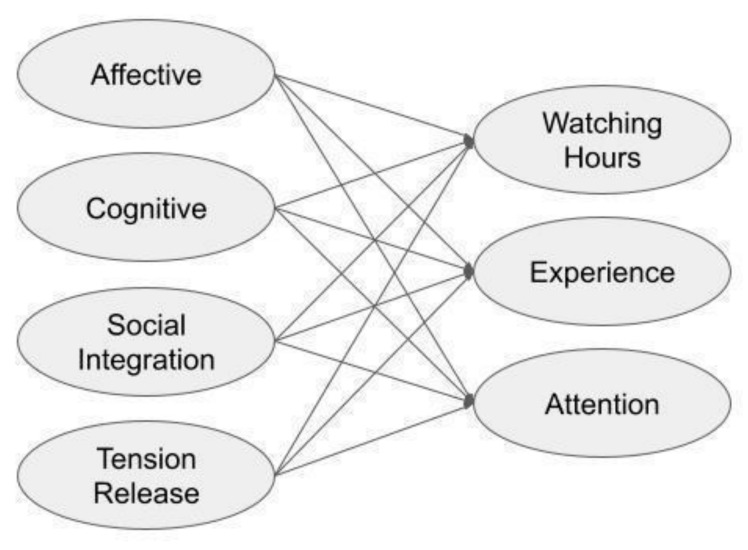
Proposed model.

**Table 1 behavsci-14-00313-t001:** Result of confirmatory factor analysis.

Item	Questionnaires	Loadings
Affective (AFF)		
1	I find watching sports (esports) to be enjoyable.	0.823
2	Watching sports (esports) is exciting.	0.742
3	I have fun watching sports (esports).	0.886
4	Watching sports (esports) is entertaining.	0.845
Cognitive (COG)		
1	By watching sports (esports), I can better decide how to play it than in the past.	0.788
2	By watching sports (esports), I can be better informed about it.	0.877
3	By watching sports (esports), I am better informed about new strategies.	0.715
Social (SOC)		
1	Watching sports (esports) makes me feel less lonely.	0.631
2	While watching sports (esports), I feel connected to the other fans.	0.910
3	While watching sports (esports), I feel a good bond with other fans of it.	0.922
Tension (TEN)		
1	While watching sports (esports), I can forget about school, work, or other things.	0.794
2	While watching sports (esports), I can get away from the rest of my family or others.	0.810
3	While watching sports (esports), I can get away from what I am doing.	0.874
Experience (EXP)		
1	I could imagine myself in the scenes I was watching.	0.720
2	I was mentally involved in the games I was watching.	0.882
Attention (ATT)		
1	I continued thinking about the games I was watching after viewing.	0.718
2	While viewing, my mind concentrated.	0.810
x^2^ = 428.827 (df = 115, *p* < 0.001), CFI = 0.946, TLI = 0.928, RMSEA = 0.070, SRMR = 0.063		

**Table 2 behavsci-14-00313-t002:** Correlation table, validity, and reliability.

	CR	α	AVE	1	2	3	4	5	6
AFF	0.889	0.888	0.667	0.817					
COG	0.834	0.828	0.629	0.650	0.793				
SOC	0.877	0.852	0.711	0.409	0.499	0.843			
TEN	0.864	0.863	0.680	0.290	0.267	0.546	0.824		
EXP	0.771	0.770	0.629	0.404	0.425	0.662	0.583	0.793	
ATT	0.724	0.720	0.568	0.531	0.535	0.641	0.641	0.824	0.754

Diagonal elements are the square root of the average variance extracted (√AVE).

**Table 3 behavsci-14-00313-t003:** Path analysis: overall model.

	Experience		Attention		Hours	
	*β*	*SE*	*β*	*SE*	*β*	*SE*
AFF	0.128 *	0.072	0.248 **	0.071	0.663	0.440
COG	0.103	0.088	0.237 **	0.086	0.568	0.536
SOC	0.593 **	0.087	0.346 **	0.077	0.760	0.468
TEN	0.291 **	0.049	0.357 **	0.048	0.058	0.282
x^2^ = 4280.827 (df = 115, *p* < 0.001), CFI = 0.946, TLI = 0.928, RMSEA = 0.070, SRMR = 0.063						

* significant at 5% level, ** significant at 1% level.

**Table 4 behavsci-14-00313-t004:** Comparison between esports and authentic sports viewers.

	Esports Viewers	Sports Viewers	*t*	*p*
Affective	5.681	5.871	−2.519	0.012
Cognitive	5.696	5.436	3.285	0.001
Social	4.964	4.896	0.672	0.502
Tension	4.619	4.850	−2.061	0.040

**Table 5 behavsci-14-00313-t005:** Comparison between non-players and players.

	Non-Player	Player	*t*	*p*
Affective	5.823	5.901	−0.842	0.401
Cognitive	5.233	5.667	−3.153	0.002
Social	4.653	5.051	−3.077	0.002
Tension	4.718	4.936	−1.565	0.119

**Table 6 behavsci-14-00313-t006:** Comparison of CFA indices for esports and sports viewers.

Model	x^2^	Δx^2^/df	*p*-Value
Unrestricted	570.35		
Weak	583.90	13.545/11	0.259
Strong	625.05	41.146/11	<0.001
Strict	715.09	90.044/17	<0.001

**Table 7 behavsci-14-00313-t007:** Path analysis for esports and authentic sports viewers.

	Experience		Attention		Hours	
	*β*	*SE*	*β*	*SE*	*β*	*SE*
**esports**						
AFF	−0.006	0.165	0.081	0.161	0.733	0.924
COG	0.218	0.198	0.300 *	0.183	0.400	1.101
SOC	0.624 **	0.136	0.283 **	0.124	−0.041	0.698
TEN	0.378 **	0.085	0.445 **	0.084	0.627	0.452
**Authentic**						
AFF	0.153 *	0.079	0.349 **	0.084	1.197 *	0.523
COG	0.096	0.093	0.144	0.096	0.231	0.619
SOC	0.589 *	0.103	0.418 **	0.103	1.323 *	0.623
TEN	0.208 *	0.061	0.316 **	0.061	0.346	0.369

* significant at 5% level, ** significant at 1% level.

**Table 8 behavsci-14-00313-t008:** Comparison of CFA models for authentic sports viewers.

Model	x^2^	Δx^2^/df	*p*-Value
Unrestricted	494.95		
Weak	506.30	11.347/11	0.415
Strong	538.69	32.396/11	<0.001
Strict	578.66	39.965/17	<0.001

**Table 9 behavsci-14-00313-t009:** Path analysis for non-players and players.

	Experience		Attention		Hours	
	*β*	*SE*	*β*	*SE*	*β*	*SE*
**Non-play**						
AFF	0.387 **	0.142	0.499 **	0.161	10.818 *	0.766
COG	0.114	0.147	0.110	0.176	−0.134	0.866
SOC	0.514 **	0.166	0.500 **	0.173	0.796	0.769
TEN	0.115	0.081	0.341 **	0.101	−0.249	0.472
**Play**						
AFF	0.012	0.093	0.232 **	0.090	0.822	0.705
COG	0.193 *	0.112	0.176 *	0.106	0.262	0.837
SOC	0.658 **	0.155	0.348 **	0.131	10.814 *	10.006
TEN	0.253 **	0.077	0.304 **	0.075	−0.423	0.550

* significant at 5% level, ** significant at 1% level.

## Data Availability

Data are contained within the article.

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
