# Peer review of "Comparison of Audience Behavior between eSports and Authentic Sports Fans"

_behavsci, 2024, doi:10.3390/bs14040313_

Round 1

Reviewer 1 Report

Comments and Suggestions for Authors

I would ask authors to provide more details about sampling and data gathering techniques and procedures as well as  information of ethical approval.

In terms of strengths, this paper extends previous research on the topic by adding a new theoretical framework. In this way, the authors gained new information and views in an already fairly well-researched field. The strength is the adaptation of the measurement instrument for the research of a specific topic as well as the verification of the metric characteristics which have been shown to be very good. This gave the scientific community another reliable tool, or rather a modification of an already used tool.

On the other hand, I'm not sure about the representativeness and relevance of the sample, that's why I advised to elaborate a little more on how the sample was chosen. Also, the validity of its application depends on the method of data collection, and this information could be elaborated in more detail.

Finally, although the theoretical framework is expanded and some new relationships are shown, the question is how relevant the obtained results are. It seems pretty average to me, which isn't necessarily a bad thing. I advise the authors to discuss the intensity of the differences, i.e. the effects, in addition to the observed differences.

Author Response

Dear Reviewer,

Thank you very much for your contributions to our manuscript. We appreciate all of your insightful and detailed comments. We are encouraged by the reviewers’ general recognition of the quality of this manuscript.

We thank the editor for providing us with this valuable opportunity to reflect on and enhance the manuscript.

We have responded to each and every comment, and have made substantial revisions to the manuscript to address the issues raised by reviewers. Changes to the original manuscript are highlighted in yellow within the revised manuscript. In addition, in our responses to reviewer comments we refer to page numbers and lines based on the Word version of the manuscript; therefore the PDF version available to reviewers may show different page or line numbers.

Reviewer 2 Report

Comments and Suggestions for Authors

The article addresses a very interesting and quite innovative topic: the Comparison of Audience Behavior between eSports and Authentic Sports Fans. Particularly intriguing is the hypothesis positing that the cognitive needs of esports viewers are higher than those of authentic sports viewers. However, I wonder if it can be stated, as it is written in the article, that all esports viewers also play games themselves. Are we certain that every single one does?

Another issue, among the 556 participants, 223 were esports audiences, and 333 were authentic sports audiences. Can this division be so unequivocal? Isn't it also the case that most esports fans also enjoy traditional sports?

Regarding the exclusion of 203 insincere answers, how were such responses determined to be insincere? Please explain.

It seems to me that in the conclusion section, there should be a brief summary of the entire article and its findings, not just an indication of the study's limitations. Overall, I find the article interesting, methodologically sound.

Author Response

Dear Reviewer,

Thank you very much for your contributions to our manuscript. We appreciate all of your insightful and detailed comments. We are encouraged by the reviewers’ general recognition of the quality of this manuscript.

We thank the editor for providing us with this valuable opportunity to reflect on and enhance the manuscript.

We have responded to each and every comment, and have made substantial revisions to the manuscript to address the issues raised by reviewers. Changes to the original manuscript are highlighted in green within the revised manuscript. In addition, in our responses to reviewer comments we refer to page numbers and lines based on the Word version of the manuscript; therefore the PDF version available to reviewers may show different page or line numbers.
